# Resin-Rich Volumes (RRV) and the Performance of Fibre-Reinforced Composites: A Review

**Amjed Saleh Mahmood** [1,2], **John Summerscales** [1,*]  and **Malcolm Neil James** [1,3]

1   Materials and Structures (MAST) Research Group, School of Engineering, Computing and Mathematics (SECaM), Reynolds Building, University of Plymouth, Plymouth PL4 8AA, UK; dramjed78@gmail.com (A.S.M.); M.James@plymouth.ac.uk (M.N.J.)
2   Electromechanical Engineering Department, University of Samarra, Samarra 34010, Salah al-Deen, Iraq
3   Department of Mechanical Engineering, Summerstrand Campus (North), Nelson Mandela Metropolitan University, Port Elizabeth 6031, South Africa
*   Correspondence: jsummerscales@plymouth.ac.uk; Tel.: +44-1752-5-86150

**Abstract:** This review considers the influence of resin-rich volumes (RRV) on the static and dynamic mechanical and physical behaviour of fibre-reinforced composites. The formation, shape and size, and measurement of RRV in composites, depending upon different fabric architectures and manufacturing processes, is discussed. The majority of studies show a negative effect of RRV on the mechanical behaviour of composite materials. The main factors that cause RRV are (a) the clustering of fibres as bundles in textiles, (b) the stacking sequence, (c) the consolidation characteristics of the reinforcement, (d) the resin flow characteristics as a function of temperature, and (e) the composite manufacturing process and cure cycle. RRV are stress concentrations that lead to a disproportionate decrease in composite strength. Those who are considering moving from autoclave consolidation to out-of-autoclave (OOA) processes should be cautious of the potential effects of this change.

**Keywords:** composites; defects; fabric reinforcement; resin-rich volumes; voids

## 1. Introduction

Recent decades have seen increasingly rapid advances in the field of fibre-reinforced composites. Pang and Bond [1] stated "that over 20 million tons [of fibre reinforced composites] are now produced every year for a variety of aerospace and other applications". However, JEC Observer [2], likely using a limited definition, suggested the global composite market in 2020 to be just 10 million tonnes. The UK Composites Strategy 2016 [3] forecasts a tripling of the market between 2016–2030. The most significant reasons for this increasing demand for high-performance materials are that composites have a high modulus and/or strength-to-weight ratio, improved fatigue performance, high toughness, and durability in harsh environments.

The aerospace industry is seeking cost reductions with out-of-autoclave prepreg (and possibly infusion) as target manufacturing routes. However, for any given reinforcement fabric, the achievable fibre volume fraction will decrease with the change from autoclave pressure (typically 5–7 bar) to a vacuum-only (<1 bar) consolidation processes. The power-law compressibility characteristics of the reinforcement could lead to a large increase in the resin fraction (a 10% decrease in fibre volume fraction is a 10% increase in the matrix volume fraction) arising from the adoption of out-of-autoclave (OOA) processes. This resin will cluster within and between the reinforcement layers.

Many authors refer to resin-rich areas (RRA), resin-rich pockets (RRP), resin-rich regions (RRR) or resin-rich zones (RRZ), by implying 2D features as a consequence of image analysis using polished microscopical sections. The features are in reality three-dimensional; therefore, this review will use the technically correct and unambiguously 3D term resin-rich volumes (RRV).

For a specific reinforcement in a constant-thickness laminate, as the number of plies and fibre volume fraction increases, there will inevitably be a decrease in the matrix fraction with a reduced number, individual volume and total volume of RRV. The matrix fraction and RRV will be highest in spray-up and hand-lamination (no consolidation pressure), less common in OOA vacuum-bag only (VBO) and liquid composite moulding (LCM), and lowest in autoclave and compression moulding processes (Figure 1). This review considers the influence of resin-rich volumes (RRV) on the mechanical and physical performance of composites.

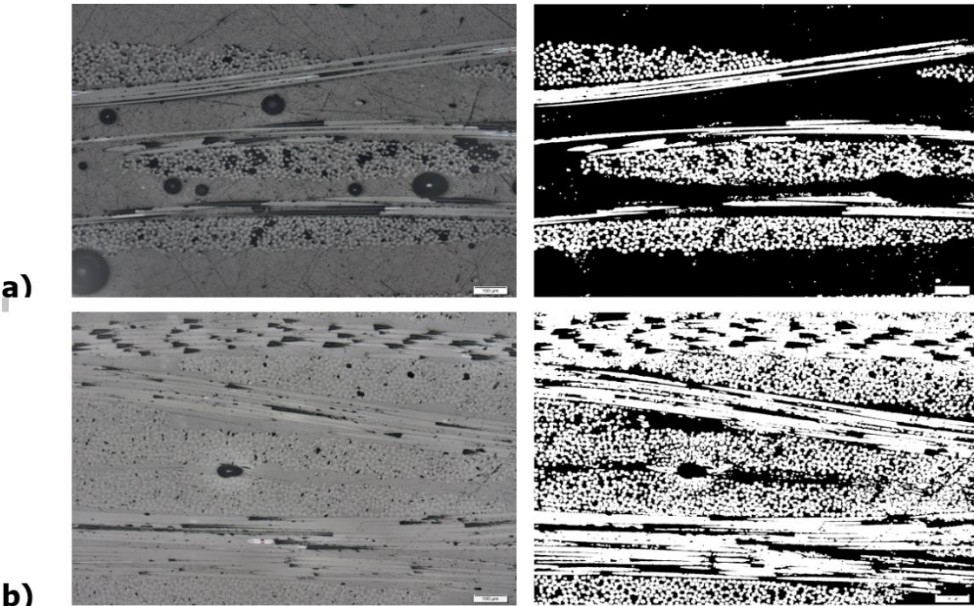

**Figure 1.** Typical micrographs (original (**left**) and binary (**right**)) of the same reinforcement with resin-rich volumes (RRV) (black in the image) with (**a**) no consolidation pressure in a hand-lamination composite (FVF = 27%) and (**b**) for vacuum-infused composites (FVF = 66%).

This review considers both experimental and numerical published research for their relevance to the following questions:

- What is the influence of processing parameters on the formation, shape and size of RRV in composites with different fabric architectures?
- What procedures and techniques can be used to measure RRV?
- How do RRV affect the behaviour of fibre-reinforced composite materials?
- What is the correlation between RRV and the space available for voids?

Most studies mention RRV as a sideline in the context of different processing parameters. To the best of our knowledge, there are no papers where the primary focus has been on controlling the size, shape and distribution of RRV within the laminate. Using fibre volume fraction (Vf) alone to calculate the mechanical properties (especially strength) of the laminates does not account for the effects of different fibre architectures. Why do composites manufactured with similar manufacturing parameters and different fibre architectures have different mechanical properties? Finding a correlation between RRV and the properties of the fibre-reinforced laminates could improve the performance of the composites.

## 2. Production of Composites and Consequent RRV

This section considers the influence of processing parameters on the formation, shape and size of RRV in four parts, dependent on the fibre architectures: (i) aligned and cross/angle-plied unidirectional (UD) composites, (ii) two-dimensional (2D) woven fabric, (iii) three dimensional (3D) woven fabric, (iv) stitched non-crimp fabric (NCF). Conclusions regarding the composites' manufacture and RRV are then presented. Indicative fibre volume fractions are presented in Table 1.

**Table 1.** Indicative fibre volume fractions for different reinforcements as a function of pressure.

| Reinforcement Form | No Consolidation (Zero Pressure) | Vacuum-Only (1 Bar) | Autoclave (5–7 Bar) |
|---|---|---|---|
| Random 2D | 10% | 20% | 30% |
| Woven 3D | 30% | 40% | 55% |
| Woven 2D | 35% | 45% | 65% |
| Unidirectional 1D | 50% | 60% | 80% |

*2.1. Aligned Unidirectional (UD), Cross-Plied (XP), and Angle-Plied (AP) UD Laminae*

In UD (all fibres aligned parallel, although some definitions permit up to 30% fibres at 90°), XP (fibres at 0° and 90°) and AP (fibres normally at −45°, 0°, +45° and 90° to give quasi-isotropic properties) composites with unidirectional laminae, the RRV are caused by different processing parameters which are related. Reducing compaction/consolidation pressure will reduce the fibre volume fraction (FVF) and resin flow and increase the RRV and voids [4]. Furthermore, the pore spaces in the dry reinforcement are filled with resin during the flow, consolidation and cure processes to create the RRV [5]. The resin flow in some consolidation processes can displace fibres and create gaps that fill with resin to form the RRV [6]. Gaps are also created due to the clustering of fibres to create space that then becomes the RRV [7]. External pressure compressed any voids within the matrix and could disperse the gasses and vapours in the matrix into the free volume between the molecules of the polymer to produce a void-free matrix.

Sudarisman and Davies [4] prepared unidirectional carbon fibre by pre-impregnating the reinforcement with a solution of epoxy resin and hardener in acetone. High solute levels lead to FVF of 46.5%, whereas lower solute levels produced consistent FVF around 52.7%. UD composite plates manufactured and cured under vacuum in an autoclave produced low-FVF composites with "matrix-rich regions between the individual prepreg layers", poor plate flatness, poor bonding between the layers, and large voids in the RRV.

*2.2. Two-Dimensional (2D) Woven Fabric*

The difference between the processing parameters, which create RRV in 2D-woven fabric (fabric architectures shown in the Figure in Section 3.1) and UD fabric, are related to fabric architecture. Increasing the manufacturing pressure, as mentioned in Section 2.1, reduces RRV, especially with autoclave consolidation [8,9], due to the nesting of fibres. RRV depend on the design of the product, fibre layout or the stacking sequence. Costa et al. [10] consolidated 8-harness, satin-weave, prepreg carbon fibre laminates in a [0°, 90°]$_{14}$ XP stacking sequence. RRV with voids were distributed at ply interfaces with cracks in RRV initiated at the void tips.

Cartié et al. [11] studied the delamination of Z-pinned (unspecified materials) unidirectional versus woven-fabric, carbon-fibre-reinforced laminates and found RRV surrounding the Z-pin. Failure micromechanisms in the test specimens were found to be dependent on the architecture of the fibre reinforcement, for example: Z-pin debonding and pullout in Mode I loading or Z-pins contributing to the crack opening displacement by pushing the crack faces apart in Mode II loading. Holmberg and Berglund [12] manufactured U-beams with 193 g/m² plain-weave carbon fibre (Brochier Injectex GB200) oriented at ±45° and found that the reinforcement pulled tight around the inner corners with consequent RRV on the outer surface.

The RRV in 12K T300 carbon fibre composites (reinforcement tape architecture not specified) with modified montmorillonite nanoclay particles in a tetrafunctional TGDDM epoxy were found to mostly occur between the laminate plies. Large RRV were correlated to a high resin viscosity when nanoparticles were included [13].

Herring and Fox [14] investigated the surface finish of carbon fibre epoxy composites with different fibre architectures (unidirectional and 2 × 2 twill) and SynSkin® surfacing

film cured using a rapid Quickstep™ procedure. High heating rates lead to low FVF, high resin content and increased surface roughness with porosity entrapment.

### 2.3. Three Dimensional (3D) Woven Fabric

In 3D woven fabric, the additional parameters (to those in UD- and 2D-woven fabric), which have a great effect on the distribution and size of RRV, include weave style and distribution of the out-of-plane binder yarn reinforcement [15,16], binder yarn path [17,18] and level of compaction [15,17].

King et al. [19] identified that, in 3D-woven fabrics, the RRV associated with the through-the-thickness binder have a direct influence on mechanical properties and a significant influence on damage development. King et al. [20] studied the influence of through-thickness binder yarn count in dry 3D carbon fibre fabrics on the prediction of FVF for fabric compaction and tow crimp. A smaller fibre count binder was expected to reduce the areal density, reduce the crimp within stuffer tows, and result in fewer RRV. Higher crimp within the through-the-thickness binders reduced the number and size of RRV.

Archer et al. [21] considered the effect of 3D weaving and consolidation on carbon fibre tows, fabrics, and composites. They identified RRV around the binder which were smaller in layer-to-layer fabrics than in angle interlock, but layer-to-layer had a higher degree of crimp in the weft filler. The warp stuffer tows had a deformed lenticular shape, elongating and conforming to the crimp of the weft filler tows. RRV were also seen between warp stuffers in the space where the binder would be on other weft layers. These RRV are an inevitable consequence with this design of fabric due to the binder pitch. Archer et al. [22] reported that weft stuffers can be pulled into the RRV between the warp stuffer stacks due to the through-the-thickness binder. The weft stuffers "pinch the edges of the warp stuffer tows distorting their shape and misaligning the warp stuffer stack". The use of binders with different tex values influences the size of the RRV.

Mahadik et al. [15] reported much larger RRV on the surface of the fabric where weaver yarns were absent above or below in order to close up the inter-yarn gaps under compaction.

### 2.4. Stitched Non-Crimp Fabric (NCF)

Mouritz et al. [23] reviewed the effect of stitching on the in-plane mechanical properties of fibre-reinforced composites and found that the stitch causes RRV. Their finding has been confirmed by other researchers of NCF [24–30], as shown in the example of Figure 2. The shape of the RRV depends on the stacking sequence of the composite laminates [28,29]. The size of the RRV are controlled by the yarn size and stacking sequence [28].

### 2.5. Conclusions Regarding Composites Manufacture and RRV

The literature clearly indicates that consolidation increases fibre volume fraction and consequently reduces the size of RRV. For a given constant reinforcement, the change from (i) hand lay-up (or spray-up), through (ii) out-of-autoclave (OOA) vacuum bag only (VBO) prepreg, RTM or RIFT, to (iii) autoclave or compression moulding decreases the size and distribution of RRV. Figure 3 illustrates the influence of processing parameters on the formation, shape and size of RRV with different fabric architectures.

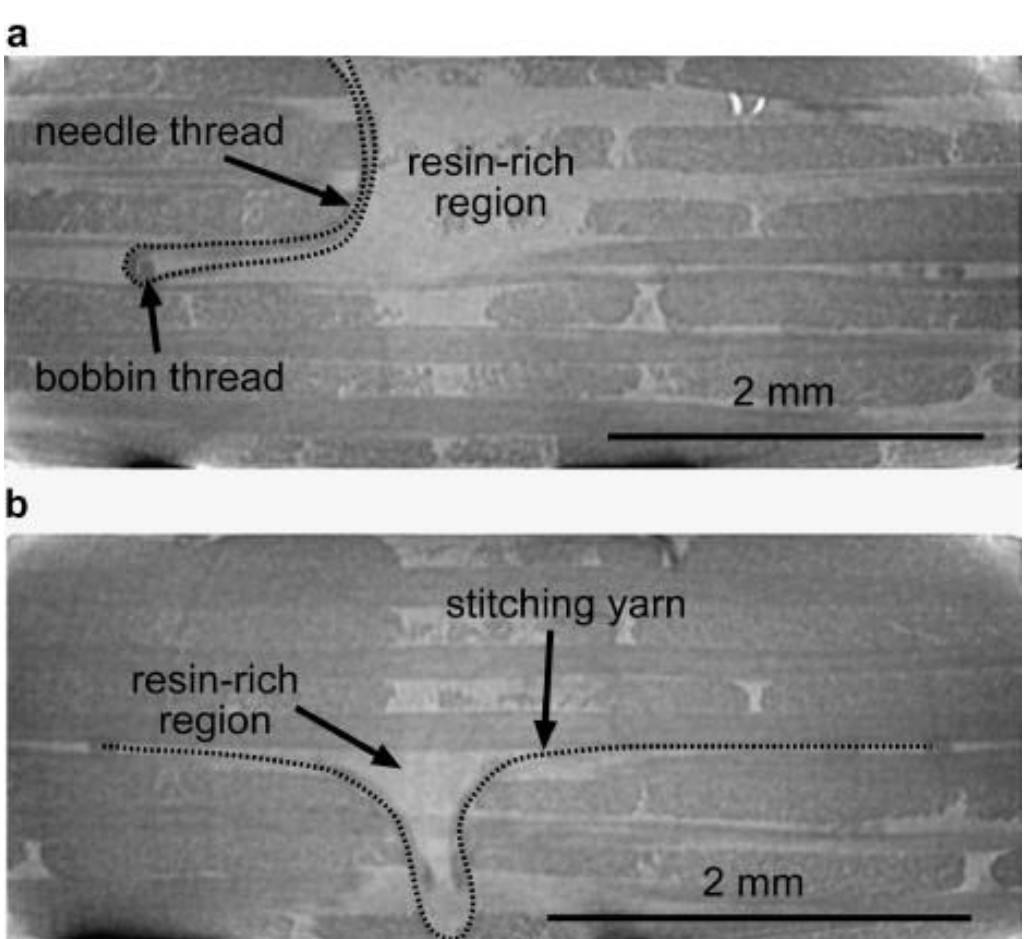

**Figure 2.** (**a**) The stitching seam direction perpendicular to the image plane and (**b**) the stitching seam laying in the image plane. (Reprinted from Composites Part A: Applied Science and Manufacturing, volume 38(7), pages 1655–1663, referenced below as Beier et al. [25] (© 2007) with permission from Elsevier, and (tbc) the authors of the paper).

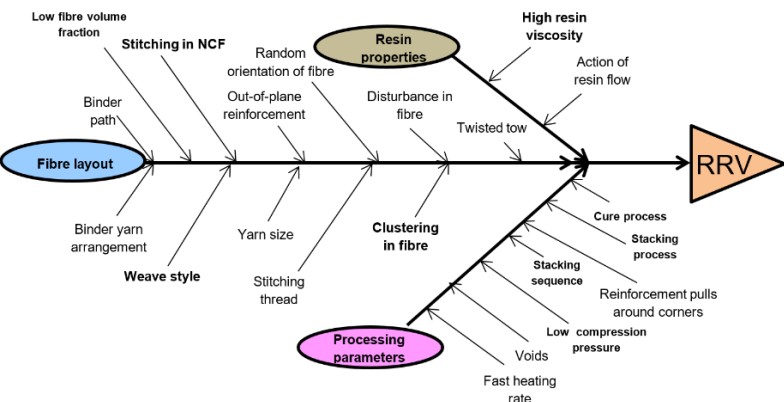

**Figure 3.** Summary of all causes of the RRV with significant influences in bold (continues in Figure 7).

## 3. Identification and Measurement of RRV

The three principal techniques used to measure the RRV in fibre-reinforced composites are (i) optical or electron microscopy, (ii) X-ray transmission, and (iii) computerised tomography (CT). High-resolution digital image correlation may extend the available techniques. Image processing and analysis is usually required.

### 3.1. Microscopic Image Analysis

Hayes [31] reviewed the optical microscopy of fibre-reinforced composites. Guild and Summerscales et al. [32–34] reviewed the use of computer-based image processing and analysis in the microscopical study of fibre-reinforced composites. Such techniques provide access to spatial information on, and the quantification of, fibre distribution in addition to volume fraction data.

Summerscales et al. [35] manufactured nine-layer unidirectional carbon-stitched fabric composite plates with cold-curing epoxy by wet lamination followed by vacuum bagging. They used different dwell times (0, 90 and 180 min) before applying vacuum pressure (0.8 bar) and found that fibre clustering and RRV were both lowest in the 90 min dwell plate.

Griffin et al. [36,37] studied commercial 2 × 2 twill carbon fibre flow-enhancement reinforcement fabrics (FERF) from Carr reinforcements woven with specially designed mesoscale architecture for RTM processes. An image analysis of optical micrographs was used to quantify the micro-/meso-structures as statistical distributions of pore space areas and perimeters. An increased flow rate was shown to be related to the presence of both modest-sized and large pore spaces in the reinforcement architecture. The pore space becomes resin-rich volumes in the composites. The authors noted that these RRV may be implicated in the premature mechanical failure of the laminate.

Pearce et al. [38] studied three Brochier (now Hexcel) carbon fibre fabrics: 5-harness satin (5HS), Injectex 5HS FERF and a 2 × 2 twill (Table 2 and Figure 4), with the same fibre type, surface treatment and FVF, to relate variations in permeability and mechanical performance to differences in the composite microstructures. The twill fabric (Figure 4a) had the smallest number of flow areas but a significant number of very large pore spaces (>0.5 mm$^2$). The satin fabric (Figure 4b) had the highest proportion of small flow areas (<0.06 mm$^2$). The Injectex (Figure 4c) had a significant number of pore space areas in the range 0.08–0.30 mm$^2$. The proportion of larger pore spaces, and the consequent fabric permeability were ranked twill > Injectex > satin, with the ranking of interlaminar shear strength (ILSS) being in reverse order. Pearce et al. [39] continued to analyse the above data set using automated image analysis, specifically a fractal dimension, to quantify the microstructure.

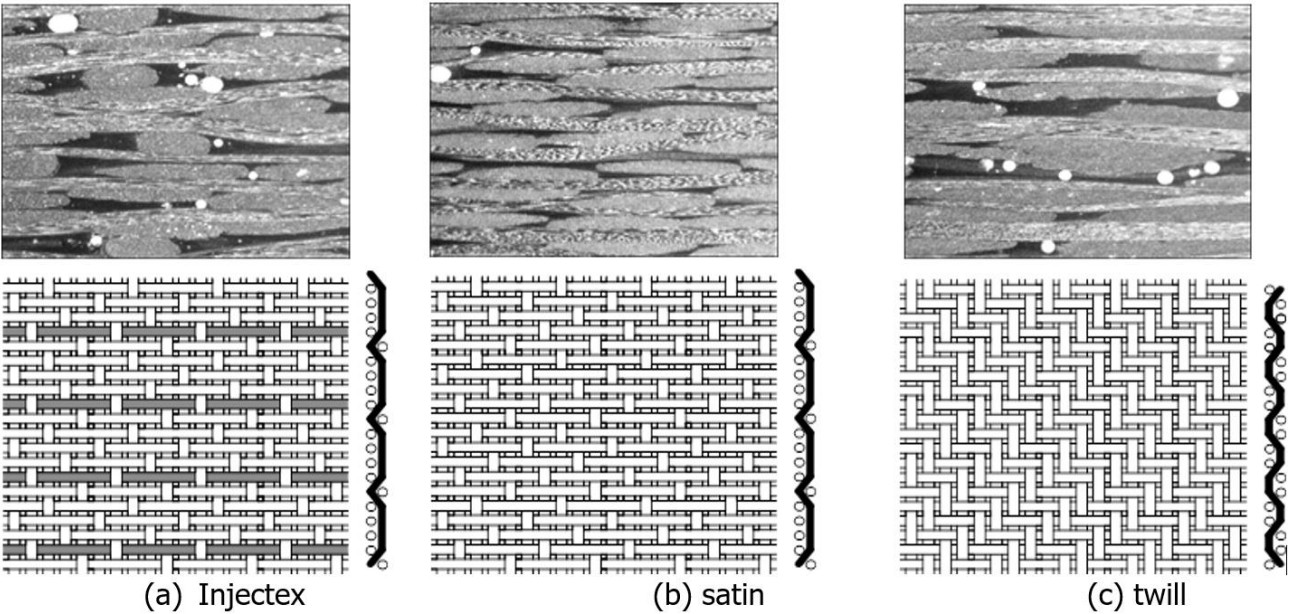

**Figure 4.** Schematics and transverse micrographs of Brochier weaves: (**a**) Injectex, (**b**) satin and (**c**) twill. Image frame is 3.6 mm × 3.0 mm showing the different RRV dependent on weave style.

**Table 2.** Fabric description, the pore space area [38], and inter-laminar shear strength (ILSS) values from ambient temperature experiments [40] for Brochier (now Hexcel)-woven carbon fibre fabrics.

| Fabric Designation | Description | % Bound Tows | Pore Space Area (mm$^2$) | Ranked by ILSS (MPa) |
|---|---|---|---|---|
| Brochier E3853 G986 6K carbon fibre fabric | 290 g/m$^2$ standard 2 × 2 twill weave | 0% | >0.5 | Lowest 45.7 ± 1.3 |
| Brochier E3833 G963 6K carbon fibre fabric | 290 g/m$^2$ Injectex 5-harness satin weave with one in five bound tows | 20% | 0.08–0.30 | Middle 53.8 ± 1.7 |
| Brochier E3795 6K carbon fibre fabric | 290 g/m$^2$ standard 5-harness satin weave | 0% | <0.06 | Highest 58.2 ± 2.6 |

Canal et al. [41] consolidated 14-layer unidirectional laminates from prepreg sheets of E-glass/MTM57 epoxy resin at 120 °C under 640 kPa autoclave pressure. Compression tests were carried out within a scanning electron microscope and a digital image correlation technique (DIC) with various magnifications (250×, 2000×, and 6000×) used to examine the differences in strains between the RRV and the fibre-rich volumes (FRV). They found that the calculated strains from DIC measurement were in a good agreement with the numerical results from a finite element model.

### 3.2. X-ray Transmission

This technique [42] projects X-rays toward the composite laminate surface, then captures the attenuated radiation on the opposite side by a detector to produce a 2D image of the laminate structure. Tan et al. [43] used X-radiography to observe in-plane matrix cracks and delamination propagation in NCF-stitched carbon fibre composites subjected to impact loading. They consolidated AP stacking sequences with either

- 32-ply [+45°/90°/−45°/0°/(0°/+45°/90°/90°/−45°/0°)$_2$]$_S$, or
- 20-ply [+45°/90°/−45°/0°/0°/+45°/90°/90°/−45°/0°]$_S$.

They conducted low-velocity, drop-weight impact tests with the drop height varied to produce different impact energies. Using zinc iodide as a radio-opaque penetrant, the radiographic images revealed that while through-thickness stitching greatly reduces delamination growth, the RRV around stitch threads act as crack initiation sites.

### 3.3. Computerised Tomography (CT)

In X-ray CT scanning [44–47], the X-ray passes through a composite laminate from different angles to create multiple images. These images can then be reconstructed to yield an arbitrary slice or a full 3D image of the laminate showing the size and distribution of the various phases including RRV or the voids. The process can be conducted at the macro-scale, but in the context of composites is normally X-ray micro-computed tomography (μCT).

Mahadik et al. [15] produced angle-interlock 3D-woven carbon fabric epoxy composites by RTM and measured the size and shape of the RRV using CT. They consolidated samples with FVF from 50% to 65% with 5% increments. They found that the RRV size (0.2 mm$^3$ to 12 mm$^3$ for different fabrics) was inversely proportional to the level of compaction. The average length of resin channels at 55% FVF was reduced by half when the FVF was increased by 5%.

Liotier et al. [28] claimed that CT could be successfully used to measure composite defects in spite of the similar X-ray absorption of carbon fibres and epoxy resin. They used stitched NCF (Hexcel NC2® Non-Crimp New Concept) carbon fibre composites as cylindrical samples with a radius of 10 mm and 2000 projections per full rotation (every 0.18°). Their paper comprehensively describes the image processing and analysis procedures. This procedure maximises the contrast between carbon fibres and epoxy, and hence improves the determination of RRV. The interlayer gaps were the main difficulty in defining the reference volume because of their differing shapes being dependent on stitch



loop geometry. RRV in multi-axial multi-ply stitched carbon composites manufactured by liquid resin infusion (LRI) averaged $3.0 \pm 0.5\%$ of the overall laminate volume.

Tan et al. [43] used CT to analyse defects and examine damage characteristics in stitched quasi-isotropic AP carbon fibre epoxy composites (as described at Section 3.2) subjected to impact loading. CT revealed that multiple cracks developed from the RRV around the closely spaced stitch threads, leading to delamination.

Fritz et al. [48] used X-ray μCT with a 1 μm voxel size to quantify morphology in autoclaved AS4/8552 and in OOA IM7/M56 aerospace-grade unidirectional-ply carbon fibre prepreg composites, identifying misplaced microfibres and periodic "tow-aligned resin pockets (TARP)" parallel to the fibre direction adjacent to the interlaminar region. They also revealed high levels of previously unreported micro-voids with an average volume of 25 $\mu m^3$.

## 4. Voids and Porosity

The pore space within a dry fabric may become RRV, and in turn may include voids (closed spaces) or porosity (inter-connected channels). Voids/porosity may include air or other volatile organic components (VOC). Subject to the resolution of the detection system and the extent of sampling, the accuracy of void volume fraction (VVF) determination is likely $\pm\ 0.5\%$ [49]. Stone and Clarke [50] reported for VVF < 1.5% that voids tend to be volatile-induced, and hence spherical with diameters in the range 5–20 μm, while for VVF > 1.5% the voids are flattened and elongated in the in-plane direction due to the limitation of space between the fibre bundles. They are also significantly larger than for voids at a lower VVF. Mayr et al. [51] have recently reported that small pores in CFRP with VVF < 1.8% often have roughly circular cross-sections and found an abrupt increase in the out-of-plane shape factors above this percentage porosity.

Judd and Wright [52] reviewed 47 papers and concluded that "although there is a considerable scatter in results (reflecting in part the difficulties of accurate void content determination) the available data show that the interlaminar shear strength of composites decreases by about 7 per cent for each 1 per cent voids up to at least the 4 per cent void content level, beyond which the rate of decrease diminishes. Other mechanical properties may be affected to a similar extent. This is true for all composites regardless of the resin, fibre or fibre surface treatment used in their fabrication". Table 1 of [52] presents a comprehensive analysis of the data. Similar findings were reported by Ghiorse [53].

Purslow [54] proposed a novel classification system for voids as the previous system was only applicable to fairly uniformly distributed voids. For example, to quote a VVF of 0.5% for a composite of generally high-quality (voids < 0.2%) but with an occasional very large void could be very misleading and potentially dangerous. He suggested that the void content should be quoted as "0 < voids < 0.2%; infrequent local voids > 0.5%". His studies suggested that when VVF < 0.5%, the voids are spherical with a diameter of 10 μm and are caused by trapped volatiles. As VVF increases, the voids decrease in number due to trapped volatiles and are replaced by large intra-tow/intra-lamina voids. The results suggested a linear relationship between VVF and void thickness, where the thickness is related to fibre diameter.

Stringer [55] used hand lay-up with 340 $g/m^2$ woven UD carbon fibre tape with a lightweight glass fibre weft in a [00/900]7 XP stacking sequence or with 472 $g/m^2$ 5-harness satin-weave carbon fibre fabric to manufacture composite laminates in epoxy resin. Increasing the epoxy viscosity to 7500–16,500 mPa.s by a dwell period before applying the vacuum pressure reduced the voids and improved the mechanical properties of the laminates. In Section 3.1, there is further evidence that well-chosen dwell periods reduce the RRV [35].

Santulli et al. [49] used a microscopic image analysis to measure the void content in twill weave E-glass/polypropylene composites. They found that coplanar voids (voids that spread over the same plane, commonly corresponding to interlayer boundaries) exist when RRV are detected between the laminae.

## 5. Evaluation of the Effects of RRV on Composite Performance

Studies normally show a negative effect of the RRV on the performance of the fibre-reinforced composites for all mechanical properties. This section is divided into four parts dealing with (i) static properties, (ii) dynamic properties, (iii) stress concentrations arising from RRV, and (iv) crack initiation and propagation.

### 5.1. Static Properties

Numerous articles describe the effect of the RRV on the mechanical properties of composites. Basford et al. [56] measured the compression and inter-laminar shear strengths of carbon fibre/epoxy composite laminates reinforced with either normal 5-harness satin (5HS) or Injectex 5HS FERF woven fabrics at constant FVF. Both strengths were found to decrease as the proportion of flow-enhancement tows increased at constant FVF (Figure 5). The twisted tow (used to enhance flow rates in Injectex) was found to cause large RRV adjacent to the tow.

Hale [16] experimentally investigated the in-plane and out-of-plane strains of 5HS-woven carbon textile composites and found that the RRV have high local/microstructural distortions that differ from theoretical models for these sites.

Gojny et al. [57] studied the effect of carbon nanotubes on the mechanical properties of glass-fibre-reinforced composites. They claimed that the RRV had a large local deformation relative to the reinforcement as would be expected given the low elastic modulus of resin.

Tzetzis and Hogg [58] investigated an infusion repair technique that could increase the toughness of carbon fibre laminates and reported a small reduction in load as presented in a load/deflection curve, due to the damage in the RRV.

Vaughan and McCarthy [59] investigated the strain distribution in HTA/6376 UD prepreg carbon fibre-epoxy laminates (stacking sequence not revealed) at micro-scale and found that transverse shear fracture bands propagated in the RRV. Liu and Liang [60] consolidated prepreg carbon/epoxy with an $[0°/-45°/+45°/90°]_s$ AP stacking sequence and reported RRV in the layers adjacent to an embedded optical fibre.

Colin de Verdiere et al. [61] showed that the RRV reduced the in-plane stiffness of tufted non-crimp fabric (NCF) composite materials by 13% relative to non-tufted reinforcement.

For natural fibre composites, Aziz and Ansell [62] investigated the effect of fibre alignment on the flexural properties of kenaf and hemp bast fibre composites. They found that the fibre alignment and the location of RRV both have a large negative effect on the flexural strength. Dhakal et al. [63] analysed the effect of water absorption on the tensile and flexural properties of hemp-fibre-reinforced composites. They found that the random orientation of fibres could produce RRV, which could then reduce the mechanical properties of the fibres.

### 5.2. Dynamic Properties (Wave Propagation, Fatigue, Impact)

The RRV have a large effect on the dynamic properties of composites. Jeong and Hsu [64] analysed wave propagation in carbon-fibre-reinforced composites and found that voids were localised in the RRV. Dyer et al. [65] studied the fatigue behaviour of one plain weave, one biaxial stitch-bonded and one quasi-isotropic stitch-bonded glass fabric-reinforced composite. Damage occurred in the RRV where cracks initiated, and then propagated. Kalam et al. [66] studied the fatigue behaviour of oil palm fruit bunch (OPFB) fibre/epoxy and carbon fibre/epoxy composites and found damage initiated in the RRV.

Azouaoui et al. [67] investigated E-glass laminates with $[(0°/90°)_2\overline{0°}]_s$ stacking sequence in impact fatigue and found that debonding and delamination cracks propagated preferentially through the interface at and in the RRV, respectively. Tan et al. [43] studied the materials described in Section 3.2 via impact loading and found that "stitches act as crack initiation sites, due to the presence of weak [RRV] around stitch threads" with densely stitched composites showing more stitch-induced matrix cracks.

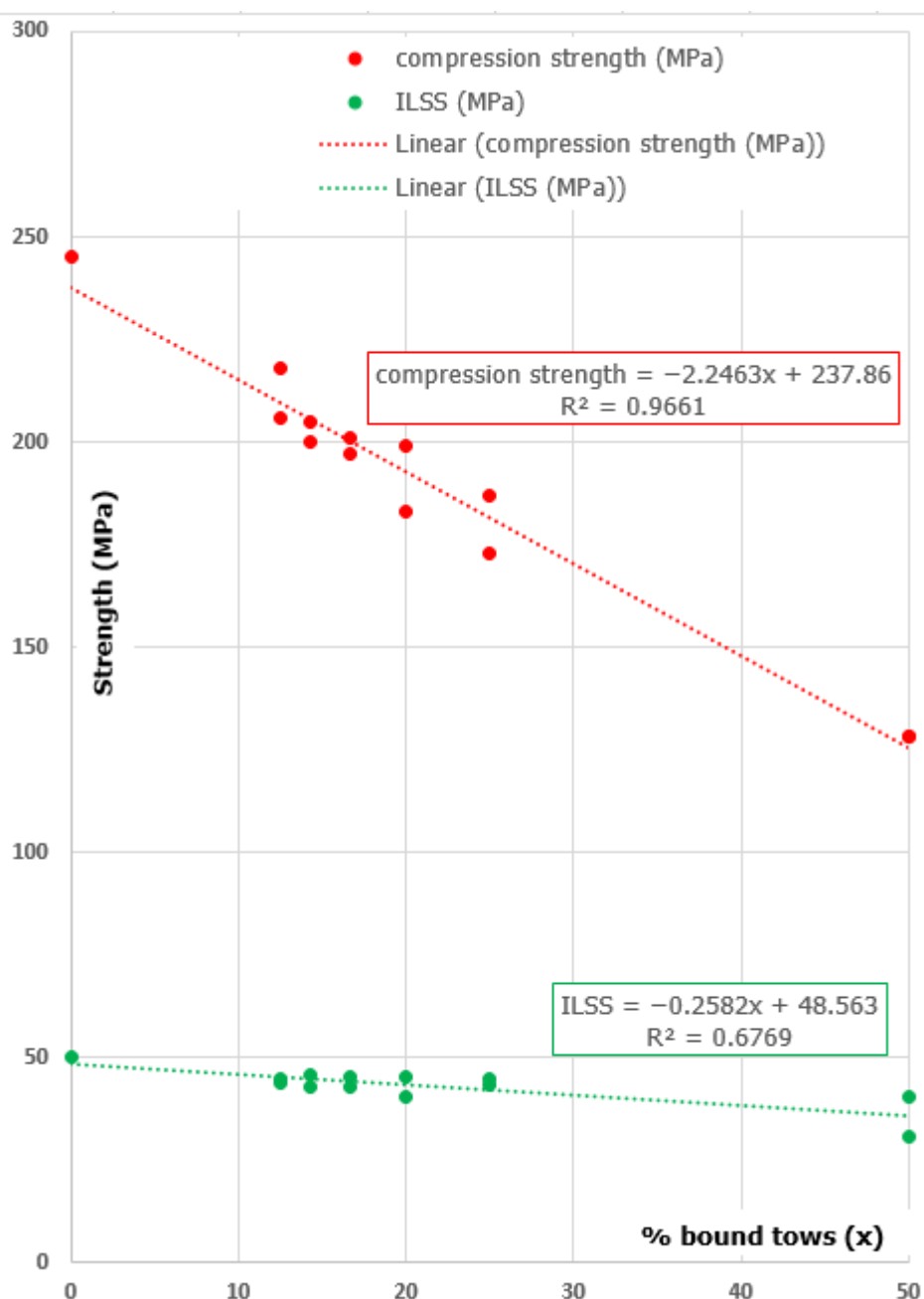

**Figure 5.** Average composite compression and interlaminar shear strengths (ILSS) vs. percentage of bound tows at constant fibre volume fraction using data from Basford et al. [56]. Paired data are for parallel or transverse tests with flow-enhancement reinforcement fabrics (FERF).

Hirano et al. [68] suggested that a resin-rich layer in graphite-epoxy laminates with different-thickness $[+45°/0°/-45°/90°]_s$ AP stacking sequences could improve impact damage resilience. Furthermore, the RRV acts as an electrical insulator and increases resistance to artificial lightning.

### 5.3. Stress Concentration

Composite laminate failure is normally initiated at stress concentrations. Mikhaluk et al. [30] claimed that the RRV could influence the mechanical behaviour of carbon/epoxy NCF laminates, and cause stress concentrations. Iarve et al. [69] analysed 3D stresses in textile composites numerically (finite element method) and experimentally (Moiré interferometry). The experiments showed very high strain sites due to the RRV

(low elastic modulus resin) between the fibre tows. Dong [70] modelled the RRV in AP glass fibre (351 g/m$^2$) composite parts using basic mechanics and observed that the RRV cause a difference in mechanical behaviour from region to region within the laminate. This study used open channel moulds, and the experimental results show a 20% difference with the model.

### 5.4. Crack Initiation and Propagation

In addition to the mechanical properties of composites, it is useful to investigate the effect of RRV on crack behaviour. As indicated above, RRV are strongly implicated in the initiation and propagation routes for crack growth [18,71–75], as shown in Figure 6. The cracks move between plies through the RRV [29,72]. The RRV act to link cracks, with more RRV leading to more micro-cracks [29] and the coherent fracture of the laminates in the RRV [76]. However, Liang et al. [77] stated that all cracks observed in their study were located at fibre/matrix interfaces, in fibre-rich volumes with almost no transverse cracks found in RRV due to the weak transverse strength of flax/epoxy.

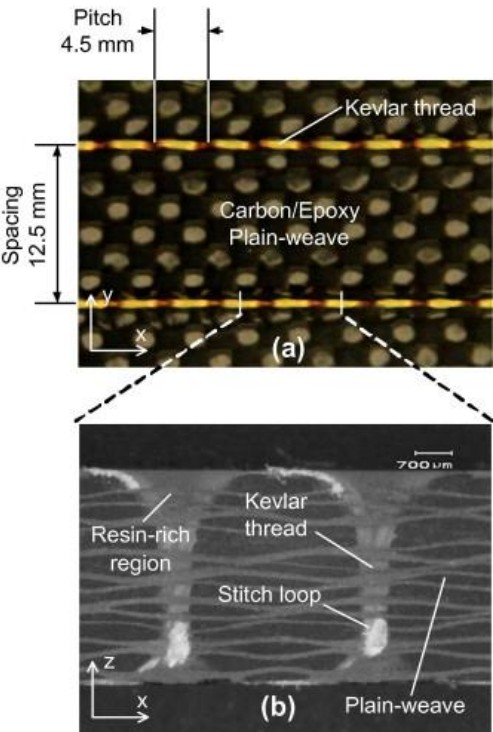

**Figure 6.** (**a**) Top view and (**b**) cross-sectional view of lock stitch pattern. (Reprinted from *Materials & Design*, volume 35, pages 563–571 referenced below as Yudhanto et al. [75] (© 2012) with permission from Elsevier, and from Arief Yudhanto).

Henne et al. [78] introduced Grilon MS phenoxy binder yarn to stabilise the preform before liquid composite moulding (LCM) processes. The yarn dissolves into the epoxy resin behind the flow front; therefore, RRV are reduced, and hence less likely to initiate microcracks.

Figure 7 summarises the effects of RRV on mechanical static and dynamic properties, their stress concentrations and the crack behaviour of fibre-reinforced composites arising from RRV.

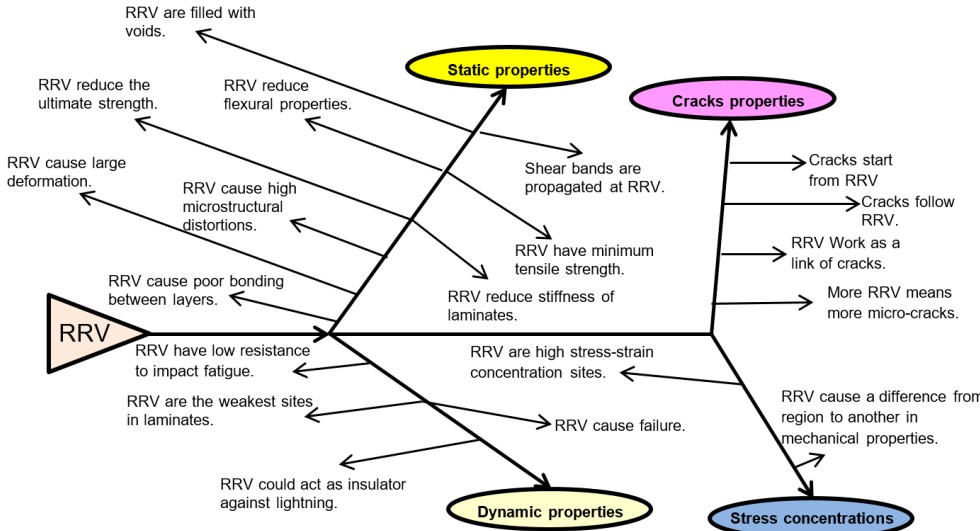

**Figure 7.** The effects of the RRV on the mechanical behaviour of fibre-reinforced composites (continued from Figure 3).

### 6. Numerical Analysis

Mesomechanics is the area that bridges the microstructure–macroproperty relationship of materials with non-continuum mechanics. Lomov et al. [79] carried out a full-field strain measurement on the surface of carbon fabric textile composites with meso-scale finite element modelling. In single-layer plain-weave carbon epoxy composites, the "lowest tensile strain value occurs not in the middle of the warp yarn but in the [RRV]". Shear concentration in the RRV between the yarns was captured by the simulation.

Heß and Himmel [80] simulated the experimental results of non-crimp fabric (NCF) carbon fibre/epoxy laminates under in-plane tension, compression and shear loading by using the finite element method. The overall mean deviation between the experiment and simulation was 8–13%. Fibre dislocation and RRV were found to reduce the laminate strength.

Ghayoor et al. [81] used a finite element analysis of representative volume elements (RVE) to model the effects of interlaminar and intralaminar RRV on the transverse stiffness and failure initiation of carbon epoxy composites. The analysis was performed using 100 different computer-simulated microstructures with geometrically varying RRV. For samples at constant FVF, the presence of intra-laminar RRV reduced the average failure initiation strains in the matrix by ~20%.

### 7. Conclusions

RRV have a significant effect on the mechanical behaviour of fibre-reinforced composites. The main techniques for measurement of RRV in fibre-reinforced composites are microscopic image analysis, X-ray transmission, and computerised tomography (CT). High-resolution digital image correlation may extend the available techniques. RRV may occupy $3.0 \pm 0.5\%$ of the total laminate volume. One study found that the individual RRV size was 0.2 mm$^3$ to 12 mm$^3$ for different fabrics. The main causes of the RRV are the fibre architecture, the resin properties and the processing parameters. Crack initiation and propagation are linked to the presence of RRV. The RRV negatively affect the static properties, the dynamic properties, the crack behaviour and stress concentration of fibre reinforced composites. One study showed that the resin-rich layer could be used as an insulator to protect against lightning.

As a consequence of the above, the authors caution against reducing the consolidation pressure and changing the fibre architectures for high-performance composite structures unless the effect of reduced FVF and the consequent RRV are fully understood for the reinforcement system being used.

### 8. Recommendations for Future Work

i. Future studies of RRV, manufacturing different laminates with similar manufacturing parameters and different fibre architectures could be useful to correlate RRV with the static and dynamic (fatigue and impact) properties of fibre-reinforced composites;

ii. A comparative study of the effect of the RRV on the mechanical properties of fibre-reinforced composites with other parameters such as FVF, crimp or stitching distribution;

iii. Modified fibre architectures could reduce the RRV and improve the performance of fibre-reinforced composites;

iv. Explore high-resolution digital image correlation techniques to visualise strain field inhomogeneity at RRV.

**Author Contributions:** Conceptualization, A.S.M. and J.S.; Methodology, A.S.M. and J.S.; Writing—Original Draft Preparation, A.S.M.; Writing—Review and Editing, A.S.M., J.S. and M.N.J.; Supervision, J.S. and M.N.J.; Project Administration, J.S. and M.N.J.; Funding Acquisition, A.S.M. All authors have read and agreed to the published version of the manuscript.

**Funding:** Amjed Saleh Mahmood is grateful to the Iraqi government and the Ministry of Higher Education and Scientific Research MOHESR for funding his doctoral studies at the University of Plymouth.

**Data Availability Statement:** This paper reviews the open scientific literature and assumes that the data behind each cited paper is in the respective institutional repositories.

**Conflicts of Interest:** The authors declare no conflict of interest.

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
