# Peer review of "Resin-Rich Volumes (RRV) and the Performance of Fibre-Reinforced Composites: A Review"

_jcs, doi:10.3390/jcs6020053_

Round 1

Reviewer 1 Report

The paper is interesting and well describes the current research potential on the topic of research and composite production. Such quality review papers are desirable for us researchers because in one place we can find an overview of the development of one segment of research that is included in a large number of published papers. I suggest improving images that are not taken from other literature such as. Figure 3, Figure 5 and Figure 7.

Author Response

Reviewer 1

The paper is interesting and well describes the current research potential on the topic of research and composite production. Such quality review papers are desirable for us researchers because in one place we can find an overview of the development of one segment of research that is included in a large number of published papers. I suggest improving images that are not taken from other literature such as. Figure 3, Figure 5 and Figure 7.

  • We thank the reviewer for the positive comments. The reviewer is too vague in respect of how the Figures might be improved (additional or revised information, different font, thicker lines?) to enable action on this comment.

Reviewer 2 Report

Dear authors,

Comments:

Page 2, paragraph 2: It would be good to give a drawing demonstrating various types of fiber architecture.  Authors allocate four. However, in the table 1, the authors use the term "form" and provide data for only three types of form.

Table 1, in the title is written: Fiber Volume Fraction As a Function of Pressure. So what level of pressure in these cases?

Subparagraph 2.1. The description of the results does not reveal the title of subparagraph.

The article often meets too general description of the results. For instance,

- page 3, line 107;

- page 4, lines 124-125;

- page 4, lines 146-147;

- page 6, lines 190-193;

Fig.3 Authors should improve the scheme in terms of selecting correct names for key factors. For example, authors include "Pores", "Reinforcement Pulls", "Stacking Process" to the process parameters, and "Action of Resin Flow" to Resin Properties.

Page 5 Lines 174-175 and 180-181. These are trivial information. It would be better to bring some experimental results.

Paragraph 3.2. It should be described in more detail how on the basis of the X-Ray Data the authors came to the conclusion described in the last sentence of this Paragraph. Maybe x-ray images need to be brought to this purpose.

The authors use Meso and Macro-Scale terms. Is this the same scale of the structure or is there a difference?

Page 10, line 324. The concentration does not relate to the process parameters.

Page 10, Lines 328-332. This paragraph does not have information about the mechanical properties.

Fig.7. The scheme in Fig. 7 is very difficult to understand that the authors wanted to demonstrate. Also occurs information that is unanswered in the text of the article. For example, the effect of Shear Band.

Author Response

Reviewer 2

Page 2, paragraph 2: It would be good to give a drawing demonstrating various types of fiber architecture.

  • We have assumed that readers of this journal, and specifically this paper would understand these fibre architecture. Given the short timescale for revision (in the first teaching week of semester and with one author suffering “covid … fever, fatigue and lack of focus” we do not have the time to produce this Figure.
  • Text has been added in the first line of §2.1 (now In UD (all fibres aligned parallel, although some definitions permit up to 30% fibres at 90°), XP (fibres at 0° and 90°) and AP (fibres normally at -45°, 0°, +45° and 90° to give quasi-isotropic properties) composites with unidirectional laminae, the RRV are caused by different processing parameters which are related. Text has been added in §2.2 (fabric architectures shown in §3.1 Figure 4), but the Figure is not brought forward as it is best positioned with the accompanying text in the current section.  Readers needing the requested diagrams will easily find then with a browser search for the image!

Authors allocate four. However, in the table 1, the authors use the term "form" and provide data for only three types of form.

  • A new row has been added to the Table for the fourth reinforcement ”form”

Table 1, in the title is written: Fiber Volume Fraction As a Function of Pressure. So what level of pressure in these cases?

  • The first row of the Table now has indicative pressures.

Subparagraph 2.1. The description of the results does not reveal the title of subparagraph.

  • Text added in the first line of §2.1 addresses this point.

The article often meets too general description of the results. For instance,

- page 3, line 107;

  • added “Z-pin debonding and pullout in Mode I loading or Z-pins contributing to the crack opening displacement by pushing the crack faces apart in Mode II loading”

- page 4, lines 124-125;

  • the sentence has been rephrased. The findings are better described in the continuation of the pargraph

- page 4, lines 146-147;

  • the key conclusion is that “the stitch causes RRV”.  2.4 expands on that topic!

- page 6, lines 190-193;

  • surely the continuation of the paragraph provides more specific information!

Fig.3 Authors should improve the scheme in terms of selecting correct names for key factors. For example, authors include "Pores", "Reinforcement Pulls", "Stacking Process" to the process parameters, and "Action of Resin Flow" to Resin Properties.

  • The authors do not use “Pores” in the Figure! Voids as a generic term includes pores (cf heading on §4 Voids and porosity)
  • The full phrase is “Reinforcement Pulls around corners”: high fibre volume fraction on inner radius and resin richness on outer surface! (now line 122)
  • “Stacking process” is the laying down of layers according to the stacking sequence!
  • “Action of resin flow” is fibre movement due to low local permeability and high resin pressure.
  • We would welcome suggestions for alternative concise phrases.

Page 5 Lines 174-175 and 180-181. These are trivial information. It would be better to bring some experimental results.

  • Lines 174-175 are further discussed in the following paragraphs
  • Lines 180-181 have been removed.

Paragraph 3.2. It should be described in more detail how on the basis of the X-Ray Data the authors came to the conclusion described in the last sentence of this Paragraph. Maybe x-ray images need to be brought to this purpose.

  • Added “They conducted low-velocity drop-weight impact tests with the drop height varied to produce different impact energies. Using zinc iodide as a radio-opaque penetrant, the radiographic images revealed that while through-thickness stitching greatly reduces delamination growth, the RRV around stitch threads act as crack initiation sites”.

The authors use Meso and Macro-Scale terms. Is this the same scale of the structure or is there a difference?

  • Meso- is normally understood to lie between micro- (up to perhaps 100 μm) and macro- (1 mm or larger). Added Mesomechanics is the area that bridges the microstructure-macroproperty relationship of materials with non-continuum mechanics.

Page 10, line 324. The concentration does not relate to the process parameters.

  • This statement (question) is not clear!

Page 10, Lines 328-332. This paragraph does not have information about the mechanical properties.

  • Paragraph moved to §2.1.

Fig.7. The scheme in Fig. 7 is very difficult to understand that the authors wanted to demonstrate. Also occurs information that is unanswered in the text of the article. For example, the effect of Shear Band.

  • The reviewer might like to suggest what is “difficult” about a multi-level list of factors expressed as a fishbone diagram?